# Inadequate lumbar protection with load weight limits based on body weight percentages: An experimental and simulation study of the weight limits set by the Japanese guidelines for preventing low back pain

Fuyuki Oyama[ID]*, Tanghuizi Du, Kazuyuki Iwakiri[ID]

Ergonomics Research Group, National Institute of Occupational Safety and Health, Japan (JNIOSH), Kawasaki, Kanagawa, Japan

* oyama@h.jniosh.johas.go.jp

**Data availability statement:** All relevant data are within the manuscript and its Supporting Information files.

## Abstract

Work-related low back pain is the most prevalent occupational disease in Japan, with 25% of cases resulting from manual handling of heavy loads. The Ministry of Health, Labor and Welfare of Japan has issued guidelines limiting load weights based on the worker's body weight percentage: 40% of the body weight for males and 24% for females. This study aimed to evaluate the effectiveness of these weight limits in preventing excessive lower back strain and identify potential shortcomings. A total of 20 healthy participants (10 males and 10 females) were included in this study. The motions of the participants while holding loads at specific positions and lifting loads from the floor were recorded using a three-dimensional motion analysis system. The compressive force on the L5-S1 intervertebral disc (IVD-CF) was estimated by simulations using inverse dynamics analysis. The body weights for the musculoskeletal models were set at 50, 70, and 90 kg for males and 40, 55, and 70 kg for females, representing the light, average, and heavy body weights of Japanese workers. The load weights were set at 40% of the body weight for males and 24% for females. Increased body weight led to higher IVD-CF in males, exceeding the safe limit of 3400 N. Additionally, light-weight males experienced excessive strain when holding loads at shin height or lifting from the floor. The IVD-CF for females was lower than that for males because of their lighter body weight and load weight. However, heavy-weight females experienced excessive strain when holding loads at low and distant positions and during lifting. These findings indicate that weight limits based on body weight do not adequately prevent excessive lower back strain, and the nature of the task should be considered to avoid lower back strain.

**Funding:** This study was supported by the National Institute of Occupational Safety and Health, Japan (N-P03-01). The funders had no role in study design, data collection and analysis, decision to publish, or preparation of the manuscript.

## Introduction

Work-related low back pain (LBP) is a significant occupational health concern worldwide. In Japan, it is the most common occupational disease, accounting for >60% of occupational disease cases requiring ≥4 days of leave [1]. One-fourth of these cases occur during the manual handling of heavy loads, highlighting it as a major risk factor for LBP [2]. The Ministry of Health, Labor and Welfare of Japan (MHLW) guidelines recommend that the load weight of manual handling should not exceed 40% of the body weight for males and 24% for females [3].

Unlike the MHLW guidelines, other international guidelines set load weight limits based on various factors, such as age, gender, and nature of the task. For instance, the International Organization for Standardization (ISO) issued ISO 11228–1:2021, which sets the maximum recommended load weight under optimal conditions based on gender and age: 25 kg for males aged 20–45 years, 20 kg for males outside this age range, 20 kg for females aged 20–45 years, and 15 kg for females outside this age range [4]. The Health and Safety Executive (HSE) in the United Kingdom has provided detailed weight limits based on the position of the hands during handling, with the heaviest allowable load weight for males being 25 kg when handled close to the body at waist height and 16 kg for females [5]. Additionally, the American Conference of Governmental Industrial Hygienists considers the frequency and duration of tasks in their guidelines [6]. The National Institute for Occupational Safety and Health (NIOSH) in the United States developed the NIOSH Lifting Equation, which sets a weight limit of 23 kg for optimal conditions and adjusts for various unfavorable conditions [7].

The MHLW guidelines raise several issues. First, they lack a maximum load weight limit because the permissible weight depends on the worker's body weight, which can lead to excessive strain on the lower back as the body weight increases. Additionally, the guidelines do not set load weight limits based on load handling positions, frequency of movements, or duration of tasks. These factors can significantly affect the strain on the lumbar intervertebral discs [8] and increase the incidence of LBP [9].

An epidemiological study on Japanese workers showed that even when load weights were within the limits set by MHLW guidelines, some workers who manually handled loads had a significantly higher incidence of LBP than those who did not handle loads [10]. Therefore, the weight limits set by the MHLW guidelines may not be adequate to prevent LBP. This may be because these limits, based only on body weight, do not prevent excessive lower back strain, resulting in compressive force on the intervertebral disc (IVD-CF) exceeding the 3400 N limit recommended by the NIOSH [7,11].

This study hypothesized that load weight limits based only on body weight percentage, as set by the MHLW guidelines, do not adequately prevent excessive strain on the lower back and are insufficient for preventing LBP. To the best of our knowledge, no previous research has examined this hypothesis from a biomechanical perspective. Therefore, this study aimed to investigate this hypothesis and identify potential shortcomings of the method of determining load weight limits based on body weight.

## Methods

### Participants

A total of 20 healthy adult participants were included in this study. Of the 20 participants, 10 were males, with a mean age of 40.8 years (standard deviation (SD) = 8.2), mean height of 169.9 cm (SD = 4.9), mean weight of 67.5 kg (SD = 12.8), and mean body mass index of 23.4 (SD = 4.7), and 10 were females, with a mean age of 40.3 years (SD = 6.0), mean height of 158.4 cm (SD = 4.7), mean weight of 60.0 kg (SD = 9.7), and mean body mass index of 23.9 (SD = 3.6). This study was approved by the ethics board of the National Institute of Occupational Safety and Health of Japan (registration ID: 2022N-1–15). Written informed consent was obtained from all participants after receiving written and oral explanations about the study's purpose and methods. The data collection period for this study was from June 1, 2023 to March 5, 2024.

### Tasks

The tasks included holding and lifting tasks. In the holding task, participants held a load at a designated holding position. The load was placed on a platform that was adjusted to the holding position, and the participants lifted it 3 cm above the surface and held it for 2 s. The holding positions included five height levels (head, chest, waist, knee, and shin) and two distance levels (near and far) from the body (Fig 1). Head height was defined as the height from the shoulder to the top of the head, chest height as the height from the elbow to the shoulder, waist height as the height from the metacarpophalangeal (MP) joint of the index finger to the elbow, knee height as the height from the middle of the lower leg to the MP joint of the index finger, and shin height as the height below the middle of the lower leg. The participants maintained a standing posture when holding the load at head, chest, or waist height, whereas they adopted a squat posture when holding the load at knee or shin height. Near distance was defined as the range within half the length of the arm when extended

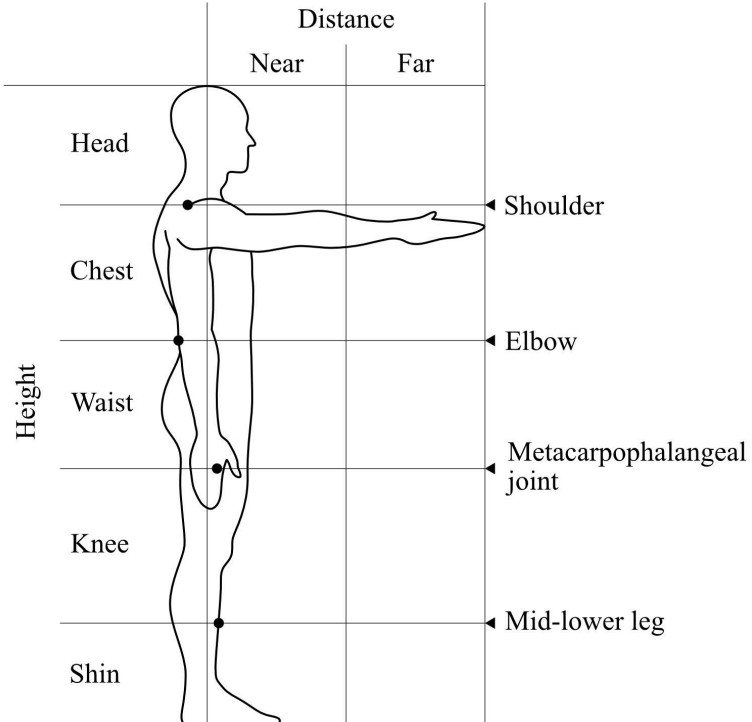

**Fig 1. Heights and distances from the body.**

forward, whereas far distance was defined as the range from half the length of the arm to the fingertips. In the lifting task, the participants lifted a load from the floor in a squat posture and raised it to the highest possible height within the range of their head level. The load was a container measuring 40.0 cm in width, 28.0 cm in depth, and 7.4 cm in height, with handles on both sides (Fig 2). The same container was used in all conditions, and the load weight was adjusted by adding metal weights. The tasks were performed on separate days to minimize fatigue.

The load weights used in the tasks were determined based on the maximum weights the participants could handle in their ordinary work. Before the motion recordings, the participants identified the heaviest load they could lift multiple times with approximately 70% of their maximum effort based on their subjective judgment. The maximum load weight was recorded for each of the 10 positions in the holding tasks. The weight options were 3, 5, 8, 10, 13, 15, 18, 20, 23, 25, and 28 kg. The maximum load weight for each position was used for the motion recordings in the holding tasks, and the maximum load weight across all 10 positions was used for the lifting task.

### Three-dimensional motion analysis

The motion of the participants during the tasks was recorded using a three-dimensional motion analysis system. The participants wore stretch fabric suits and infrared reflective markers. The markers were attached based on the Plug-in Gait model [12], with additional markers to cover blind spots, such as the chest, pelvis, wrists, and feet, totaling 54 markers. The motion was captured at a sampling rate of 100 Hz using 10 infrared cameras (Vicon Vero, Vicon Motion Systems Ltd., UK) and capture software (Vicon Nexus 2.16, Vicon Motion Systems Ltd.). Fig 3 shows examples of the captured motion data while holding the load. The markers positioned at the MP joint of the index fingers are located near the handles of the container, and this position is considered the handling position of the load.

### Estimation of lower back strain

Lower back strain was assessed using the L5-S1 IVD-CF, a key indicator of lower back strain [11]. The IVD-CF was estimated by simulations based on the three-dimensional motion data. A musculoskeletal model was applied to the motion data to calculate the muscle forces required for the movements. These muscle forces were used to determine the forces on each joint, and the maximum IVD-CF was obtained.

The analysis was performed using musculoskeletal simulation software (AnyBody 7.4, AnyBody Technology A/S, Denmark), which has been validated for accuracy [13–15] and widely used to evaluate the strain on the lumbar spine during

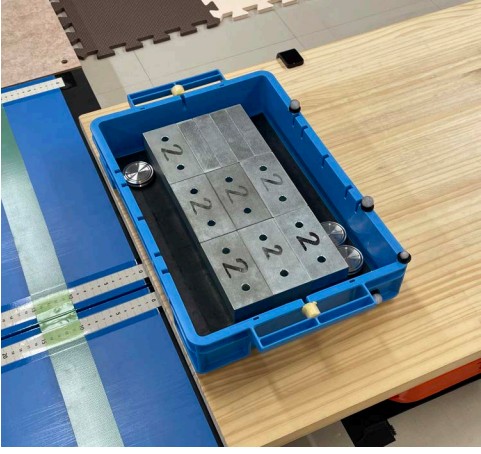

**Fig 2. Container and metal weights.** An example where the load weight is 20 kg.

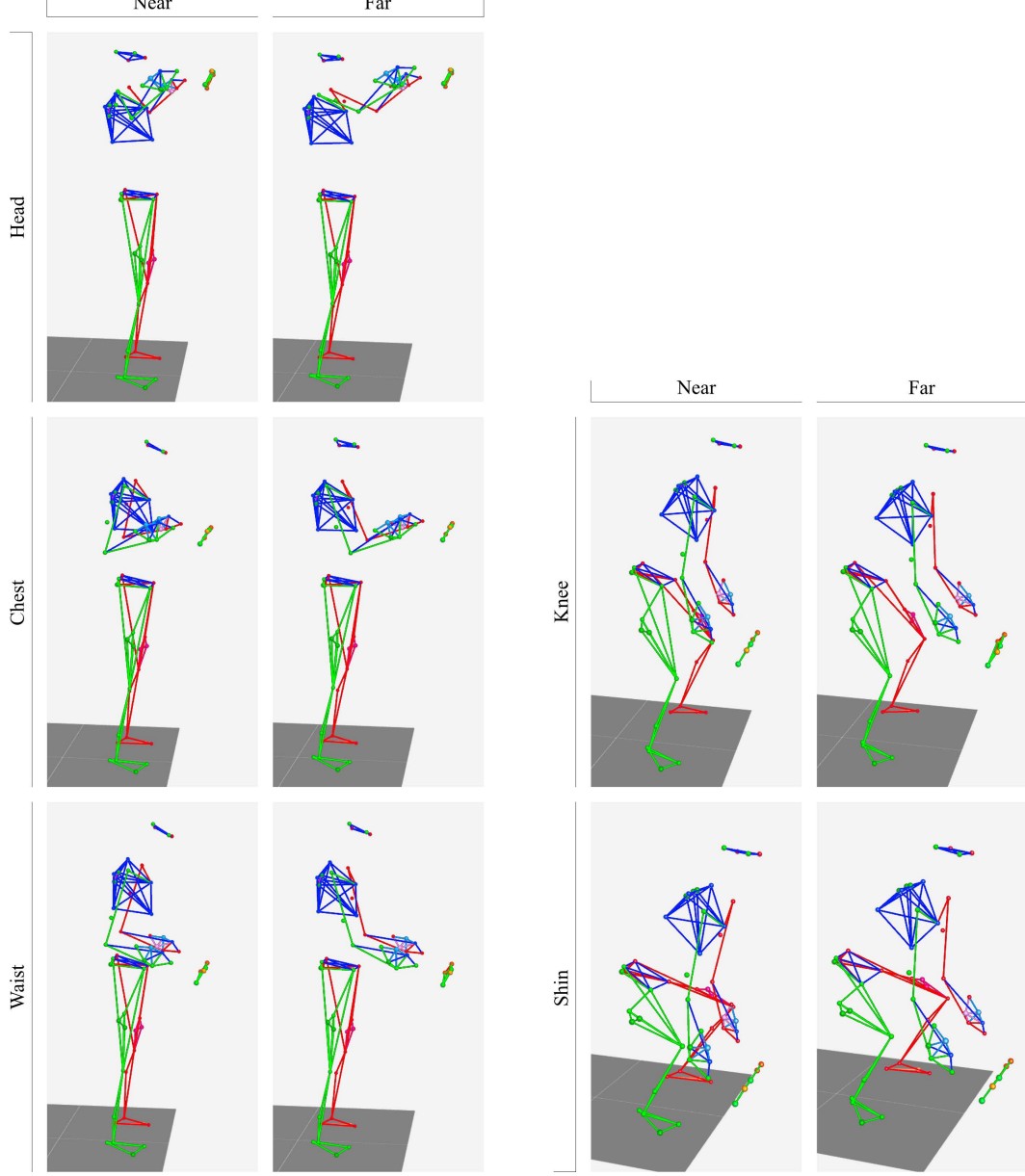

**Fig 3. Examples of the captured motion data while holding the load.** The dot at the tip of the arm indicates a marker positioned at the MP joint of the index finger.

the lifting tasks [16–18]. The skeletal model was adjusted to each participant's body to match the segment lengths. The adjusted skeletal model was fitted to the motion data for the analysis period. For the holding tasks, the analysis period was 1 s during static holding. For the lifting tasks, the analysis period was from lifting the load off the floor to placing it back on the floor. In the musculoskeletal model, the muscle forces required to support the load and body weight were calculated. From these forces, the maximum IVD-CF was determined.

In the simulations, body and load weights were manipulated according to three different weight conditions for each gender. Weight conditions were determined based on the distribution of body weights in the Japanese population [19].

For each gender, the approximate average body weight (average weight) and the average plus/minus approximately 1.67 SD (heavy weight and light weight, respectively) were included in the conditions. This ensured that body weights for more than 90% of the population were expected to fall within the range of light to heavy weights. For males, light, average, and heavy weights were set at 50, 70, and 90 kg, respectively. The load weight was 40% of the body weight: 20 kg for 50 kg, 28 kg for 70 kg, and 36 kg for 90 kg. For females, light, average, and heavy weights were set at 40, 55, and 70 kg, respectively. The load weight was 24% of the body weight: 9.6 kg for 40 kg, 13.2 kg for 55 kg, and 16.8 kg for 70 kg. Inverse dynamics analysis requires the inputs of all external forces, including ground reaction forces, to enhance the accuracy [20]. However, the actual measured ground reaction force data could not be used due to the manipulation of the body and load weights in this simulation. Therefore, the ground reaction force estimation function built into the simulation software was used for analysis without inputting the actual measured ground reaction force data.

## Statistical analysis

According to the NIOSH guidelines, IVD-CF exceeding 3400 N increases the risk of LBP [11]. Therefore, the probability of the maximum IVD-CF being below 3400 N in the estimated population distribution was calculated to confirm that the maximum IVD-CF was below 3400 N. If this probability was > 95%, the maximum IVD-CF was considered <3400 N. This is equivalent to the 95th percentile being below 3400 N. Therefore, the 95th percentile was also reported in the results.

For the holding tasks, a three-way repeated measures ANOVA was performed for each gender with weight, height, and distance as factors. For the lifting tasks, a one-way repeated measures ANOVA was performed for each gender, with weight as the factor. The maximum IVD-CF was log-transformed to improve the homogeneity of variance. If Mauchly's sphericity test was significant, the degrees of freedom were corrected using Greenhouse–Geisser epsilon. The significance level was set at 0.05. The generalized eta squared ($\eta_G^2$) was calculated to compare effect sizes. Post hoc multiple comparison tests were performed to further investigate the significant main effects and interactions identified by the ANOVA. The *p*-values were adjusted using the Bonferroni correction to account for multiple comparisons. Statistical analyses were performed using R version 4.4.1, with the ez_4.4–0 package for ANOVA and the emmeans_1.10.6 package for multiple comparison tests.

## Results

Table 1 shows the maximum weight capacities. The maximum weight capacity for males was 19.7 kg at waist height, 17.1 kg at knee height, 16.5 kg at shin height, 16.2 kg at chest height, and 9.7 kg at head height in the near distance. The weight capacity at a position in the far distance was lower than that at the same height in the near distance, decreasing by approximately 16% at head height and 30%–42% at other heights. The maximum weight capacity for females was 14.0 kg at waist height, 14.5 kg at knee height, 13.8 kg at shin height, 11.1 kg at chest height, and 7.7 kg at head height in the near distance. The weight capacity at a position in the far distance was lower than that at the same height in the near distance, decreasing by approximately 16% at head height and 23%–31% at other heights.

Table 2 shows the maximum IVD-CF for males obtained from the simulation. For light-weight males weighing 50 kg, the IVD-CF was likely to exceed 3400 N only during the holding tasks at shin height in the near distance and at shin and knee heights in the far distance and during the lifting tasks. For average-weight males weighing 70 kg, the IVD-CF was < 3400 N during the holding tasks at and above waist height in the near distance. However, it could exceed 3400 N at knee and shin heights. Additionally, the IVD-CF could exceed 3400 N during the holding tasks at all heights in the far distance. The IVD-CF could exceed 3400 N during the lifting tasks. For heavy-weight males weighing 90 kg, the IVD-CF could exceed 3400 N in all positions during the holding and lifting tasks.

Table 3 shows the maximum IVD-CF for females obtained from the simulation. For light-weight (40 kg) and average-weight (55 kg) females, no IVD-CF reached 3400 N during the tasks. However, for heavy-weight females weighing 70 kg, the IVD-CF could exceed 3400 N during the holding tasks at shin height in the far distance and during the lifting tasks.

**Table 1. Maximum weight capacity (kg) for males and females.**

| Males (n=10) | | | | | |
|---|---|---|---|---|---|
| | | **Near** | | **Far** | |
| **Task** | **Height** | **Mean** | **SD** | **Mean** | **SD** |
| Holding | Head | 9.7 | 2.8 | 8.1 | 2.3 |
| | Chest | 16.2 | 4.0 | 10.4 | 2.4 |
| | Waist | 19.7 | 4.9 | 11.5 | 3.0 |
| | Knee | 17.1 | 4.3 | 12.0 | 3.0 |
| | Shin | 16.5 | 4.0 | 11.2 | 3.2 |
| Lifting | | 23.1 | 4.4 | | |
| **Females (n=10)** | | | | | |
| | | **Near** | | **Far** | |
| **Task** | **Height** | **Mean** | **SD** | **Mean** | **SD** |
| Holding | Head | 7.7 | 2.1 | 6.5 | 1.6 |
| | Chest | 11.1 | 3.0 | 8.1 | 2.3 |
| | Waist | 14.0 | 2.6 | 9.7 | 2.1 |
| | Knee | 14.5 | 4.1 | 10.6 | 3.2 |
| | Shin | 13.8 | 3.9 | 10.6 | 3.2 |
| Lifting | | 17.1 | 4.6 | | |

A three-way repeated measures ANOVA was performed to investigate the effects of weight, height, and distance on the maximum IVD-CF during the holding tasks for males (Table 4). The results showed significant main effects with large effect sizes for weight, height, and distance. Regarding the interactions, although some $p$-values were below the significance level, the effect sizes were relatively small. The post hoc multiple comparison tests showed that as the weight increased, the maximum IVD-CF significantly increased in the near and far distances ($p < 0.001$ for all pairs). The maximum IVD-CF tended to decrease monotonically as the height increased. Under both distance conditions, the maximum IVD-CF at waist, chest, and head heights were significantly less than that at shin and knee heights ($p < 0.001$ for all pairs). Only under the near-distance condition, the maximum IVD-CF at knee height was significantly less than that at shin height ($p = 0.048$), and the maximum IVD-CF at head height was significantly less than that at waist height ($p = 0.027$). The maximum IVD-CF at the far distance was significantly greater than that at the near distance under all weight conditions ($p < 0.001$ for all pairs) and all height conditions ($p < 0.001$ for all pairs).

Similarly, for females, the ANOVA revealed significant main effects with large effect sizes for weight, height, and distance. Regarding interactions, a moderate effect size was observed for the interaction between height and distance, whereas other interactions had minute effect sizes. The post hoc multiple comparison tests showed that as the weight increased, the maximum IVD-CF significantly increased under all height conditions ($p < 0.001$ for all pairs) and distance conditions ($p < 0.001$ for all pairs). The maximum IVD-CF tended to decrease monotonically as the height increased. Under all weight conditions, significant decreases in the maximum IVD-CF were observed at waist, chest, and head heights compared with shin and knee heights ($p < 0.001$ for all pairs), at chest height compared with waist height ($p = 0.047–0.048$), and at head height compared with waist height ($p < 0.001$ for all pairs). Under both distance conditions, significant decreases were observed at waist, chest, and head heights compared with shin and knee heights ($p < 0.001$ for all pairs) and at head height compared with waist height ($p < 0.001$ for the pair under the near-distance condition; $p = 0.021$ for the pair under the far-distance condition). Furthermore, significant decreases in the maximum IVD-CF were observed at head height compared with chest height under the near-distance condition ($p = 0.001$) and at knee height compared with shin height under the far-distance condition ($p = 0.031$). The maximum IVD-CF at the far distance was significantly

**Table 2. Maximum L5-S1 IVD-CF (N) for males.**

**Light-weight males (50 kg) with a 20 kg load (*n*=10)**

| Task | Height | Near | | | | Far | | | |
|---|---|---|---|---|---|---|---|---|---|
| | | Mean | SD | P(<3400) | P95 | Mean | SD | P(<3400) | P95 |
| Holding | Head | 1785 | 157 | 1.000 | 2043 | 2462 | 254 | 1.000 | 2880 |
| | Chest | 1900 | 222 | 1.000 | 2265 | 2515 | 239 | 1.000 | 2908 |
| | Waist | 1983 | 246 | 1.000 | 2388 | 2685 | 325 | 0.986 | 3219 |
| | Knee | 2669 | 295 | 0.993 | 3154 | 3339 | 256 | 0.594 | 3760* |
| | Shin | 2933 | 328 | 0.923 | 3473* | 3530 | 313 | 0.339 | 4045* |
| Lifting | | 3243 | 440 | 0.639 | 3967* | | | | |

**Average-weight males (70 kg) with a 28 kg load (*n*=10)**

| Task | Height | Near | | | | Far | | | |
|---|---|---|---|---|---|---|---|---|---|
| | | Mean | SD | P(<3400) | P95 | Mean | SD | P(<3400) | P95 |
| Holding | Head | 2497 | 221 | 1.000 | 2860 | 3445 | 357 | 0.450 | 4033* |
| | Chest | 2657 | 312 | 0.991 | 3171 | 3519 | 337 | 0.362 | 4073* |
| | Waist | 2772 | 346 | 0.965 | 3342 | 3756 | 457 | 0.218 | 4507* |
| | Knee | 3735 | 414 | 0.209 | 4416* | 4673 | 361 | 0.000 | 5266* |
| | Shin | 4105 | 461 | 0.063 | 4863* | 4941 | 440 | 0.000 | 5664* |
| Lifting | | 4539 | 616 | 0.032 | 5552* | | | | |

**Heavy-weight males (90 kg) with a 36 kg load (*n*=10)**

| Task | Height | Near | | | | Far | | | |
|---|---|---|---|---|---|---|---|---|---|
| | | Mean | SD | P(<3400) | P95 | Mean | SD | P(<3400) | P95 |
| Holding | Head | 3207 | 284 | 0.751 | 3674* | 4425 | 460 | 0.013 | 5182* |
| | Chest | 3413 | 402 | 0.487 | 4075* | 4521 | 434 | 0.005 | 5234* |
| | Waist | 3561 | 446 | 0.359 | 4294* | 4825 | 588 | 0.008 | 5792* |
| | Knee | 4799 | 533 | 0.004 | 5675* | 6004 | 464 | 0.000 | 6767* |
| | Shin | 5274 | 592 | 0.001 | 6248* | 6348 | 565 | 0.000 | 7278* |
| Lifting | | 5832 | 792 | 0.001 | 7135* | | | | |

SD, standard deviation; IVD-CF, compressive force on the L5-S1 intervertebral disc; *P*(<3400), probability of being below 3400 N; P95, 95th percentile values.

*P95 > 3400 N.

greater than that at the near distance under all weight conditions (*p*<0.001 for all pairs) and all height conditions (*p*<0.001 for all pairs).

A one-way repeated measures ANOVA was performed using weight as the factor for the maximum IVD-CF during the lifting tasks. The results showed substantial effect sizes for the main effect of weight for both males and females. The post hoc multiple comparison tests showed that as the weight increased, the maximum IVD-CF significantly increased for both males (*p*<0.001 for all pairs) and females (*p*<0.001 for all pairs).

## Discussion

This study investigated the hypothesis that setting load weight limits at 40% of body weight for males or 24% of body weight for females would not keep the strain on the lower back within the acceptable limit of <3400 N of the IVD-CF and identified potential shortcomings of the body weight-based method of determining weight limits. The results showed that even when the load weights conformed to these limits, handling them could place excessive lower back strain under certain conditions. Additionally, the analysis revealed that the strain significantly increased when the body weight increased,

**Table 3. Maximum L5-S1 IVD-CF (N) for females.**

**Light-weight females (40 kg) with a 9.6 kg load (*n* = 10)**

| Task | Height | Near | | | | Far | | | |
|------|--------|------|-----|--------|-----|------|-----|--------|-----|
| | | Mean | SD | P(<3400) | P95 | Mean | SD | P(<3400) | P95 |
| Holding | Head | 837 | 142 | 1.000 | 1071 | 1179 | 96 | 1.000 | 1337 |
| | Chest | 964 | 101 | 1.000 | 1130 | 1207 | 105 | 1.000 | 1379 |
| | Waist | 1068 | 72 | 1.000 | 1187 | 1324 | 86 | 1.000 | 1465 |
| | Knee | 1355 | 120 | 1.000 | 1553 | 1625 | 99 | 1.000 | 1787 |
| | Shin | 1449 | 93 | 1.000 | 1601 | 1819 | 104 | 1.000 | 1990 |
| Lifting | | 1775 | 131 | 1.000 | 1991 | | | | |

**Average-weight females (55 kg) with a 13.2 kg load (*n* = 10)**

| Task | Height | Near | | | | Far | | | |
|------|--------|------|-----|--------|-----|------|-----|--------|-----|
| | | Mean | SD | P(<3400) | P95 | Mean | SD | P(<3400) | P95 |
| Holding | Head | 1147 | 196 | 1.000 | 1469 | 1617 | 133 | 1.000 | 1835 |
| | Chest | 1322 | 139 | 1.000 | 1551 | 1657 | 145 | 1.000 | 1895 |
| | Waist | 1464 | 100 | 1.000 | 1628 | 1817 | 115 | 1.000 | 2007 |
| | Knee | 1860 | 166 | 1.000 | 2133 | 2232 | 135 | 1.000 | 2454 |
| | Shin | 1989 | 128 | 1.000 | 2199 | 2498 | 144 | 1.000 | 2734 |
| Lifting | | 2438 | 180 | 1.000 | 2734 | | | | |

**Heavy-weight females (70 kg) with a 16.8 kg load (*n* = 10)**

| Task | Height | Near | | | | Far | | | |
|------|--------|------|-----|--------|-----|------|-----|--------|-----|
| | | Mean | SD | P(<3400) | P95 | Mean | SD | P(<3400) | P95 |
| Holding | Head | 1456 | 250 | 1.000 | 1867 | 2054 | 170 | 1.000 | 2333 |
| | Chest | 1679 | 177 | 1.000 | 1970 | 2105 | 185 | 1.000 | 2409 |
| | Waist | 1859 | 127 | 1.000 | 2069 | 2309 | 145 | 1.000 | 2548 |
| | Knee | 2364 | 212 | 1.000 | 2712 | 2838 | 172 | 0.999 | 3120 |
| | Shin | 2528 | 162 | 1.000 | 2795 | 3176 | 183 | 0.889 | 3478* |
| Lifting | | 3100 | 229 | 0.905 | 3477* | | | | |

SD, standard deviation; IVD-CF, compressive force on the L5-S1 intervertebral disc; *P*(<3400), probability of being below 3400 N; P95, 95th percentile values.

*P95 > 3400 N.

the holding height was below the waist level, and the load was handled at a far distance. These findings confirm the hypothesis and reveal the shortcomings of body weight-based methods for determining weight limits.

The results showed that body weight had a significant effect. Since weight limits are set at 40% or 24% of workers' body weight, this effect may be due to the combined effects of increased load and body weights. As the body and load weights increase, the number of handling positions that keep the lower back strain within the acceptable limit decreases. Consequently, male workers with higher body weights have few or no posture and position choices to handle a load of 40% of their body weight without placing undue stress on their lower back. For females, handling a load of 24% of their body weight generally keeps strain within the acceptable limit, likely due to their lighter body weights and the smaller percentage. However, excessive strain can still occur in heavier females.

The results showed that the position of handling and the presence of movement had a significant effect on lower back strain. Particularly, handling loads at lower heights or far from the body resulted in greater strain on the lower back. These findings are consistent with those of previous studies, which showed that handling loads at a lower height and away from the body increases lower back strain [8,21]. Additionally, lifting from the floor tended to impose greater strain than static

**Table 4. ANOVA summary tables.**

**Males**

**Holding task**

| Effect | $df_{\text{Effect}}$ | $df_{\text{Error}}$ | F | p | $\eta^2_G$ |
|---|---|---|---|---|---|
| Weight | 1.037 | 9.336 | 3434495.615 | <0.001** | 0.851 |
| Height | 1.644 | 14.794 | 88.096 | <0.001** | 0.750 |
| Distance | 1.000 | 9.000 | 204.856 | <0.001** | 0.634 |
| Weight × height | 2.406 | 21.651 | 2.858 | 0.071 | <0.001 |
| Weight × distance | 1.094 | 9.844 | 7.020 | 0.023* | <0.001 |
| Height × distance | 2.199 | 19.791 | 9.662 | <0.001** | 0.058 |
| Weight × height × distance | 2.361 | 21.246 | 0.774 | 0.493 | <0.001 |

**Lifting task**

| Effect | $df_{\text{Effect}}$ | $df_{\text{Error}}$ | F | p | $\eta^2_G$ |
|---|---|---|---|---|---|
| Weight | 1.045 | 9.409 | 3815990.308 | <0.001** | 0.767 |

**Females**

**Holding task**

| Effect | $df_{\text{Effect}}$ | $df_{\text{Error}}$ | F | p | $\eta^2_G$ |
|---|---|---|---|---|---|
| Weight | 1.014 | 9.122 | 1416650.368 | <0.001** | 0.873 |
| Height | 1.664 | 14.975 | 82.020 | <0.001** | 0.826 |
| Distance | 1.000 | 9.000 | 108.476 | <0.001** | 0.660 |
| Weight × height | 2.120 | 19.080 | 5.902 | 0.009** | <0.001 |
| Weight × distance | 1.093 | 9.839 | 43.650 | <0.001** | <0.001 |
| Height × distance | 4.000 | 36.000 | 9.533 | <0.001** | 0.101 |
| Weight × height × distance | 2.531 | 22.781 | 1.062 | 0.376 | <0.001 |

**Lifting task**

| Effect | $df_{\text{Effect}}$ | $df_{\text{Error}}$ | F | p | $\eta^2_G$ |
|---|---|---|---|---|---|
| Weight | 1.036 | 9.323 | 1811778.569 | <0.001** | 0.918 |

$df_{\text{Effect}}$, corrected degrees of freedom for the effect; $df_{\text{Error}}$, corrected degrees of freedom for the error; $\eta^2_G$, generalized eta squared.

*$p < 0.05$;

**$p < 0.01$.

holding at shin height in the near distance. This is because the movement during lifting increases the forces exerted on the lumbar discs [22]. Therefore, even if the strain at near and higher positions is within the acceptable limit, the strain at lower heights, far from the body, or during lifting can exceed the limit.

The results of this study showed several shortcomings of the method of determining weight limits based only on body weight. This method allows heavier workers to handle heavier loads. For instance, 90 kg male workers can handle a load of 36 kg, which is 40% of their body weight. This limit is significantly higher than the highest limits set by other organizations for male workers (e.g., 25 kg by ISO and HSE and 23 kg by NIOSH) and results in excessive strain on the lower back, which is confirmed in this study. Reducing the percentage from 40% might help mitigate the strain on the lower back. However, although this would make the weight limit for heavier workers more appropriate, it would make the weight that lighter workers can handle impractically low. This indicates that adjusting the percentage is inadequate for revising weight limits. Additionally, this will not solve the fundamental issue that strain increases as body weight increases. Therefore, weight limits should not be increased in proportion to body weight. Another shortcoming is that the nature of the task, such as the position of handling and the presence of movement, was not considered for the weight limits. This study

confirmed that even when light-weight males handled a 20 kg load, the strain exceeded the acceptable limit when holding at low and distant positions or lifting from the floor. These findings indicate that the nature of the task considerably affects the strain on the lower back. Therefore, the nature of the task should be considered when determining the weight limits. Furthermore, movement frequency and task duration were not considered when determining weight limits. Although these factors were uninvestigated in this study, they should be included in the guidelines considering their relationships with the incidence of LBP [9].

Instead of relying on body weight to determine weight limits, future guidelines should establish weight limits based on various factors, such as gender, age, task nature, movement frequency, and task duration. This method of determining weight limits involves the use of the equation from the ISO guidelines. In this equation, the weight limit under optimal conditions is predetermined as the maximum value. Multipliers based on various factors, including deviation from optimal height, vertical travel distance, horizontal distance from the body, twisting angle, difficulty in handling the load, task frequency, and duration, can be applied to the maximum weight limit to calculate the weight limit for a specific task. These multipliers range from 0 to 1 and serve to reduce the weight limit. The appropriate multipliers for these factors can be obtained from reference tables prepared for each factor. Although this method allows for a detailed assessment of tasks, it can be challenging owing to the complexity of reading and calculating the multipliers. Therefore, this method is more suited for detailed risk assessments rather than for determining on-site weight limits in a straightforward manner. When applying this method to the Japanese guidelines, investigating whether the weight values and multipliers from the ISO guidelines are appropriate for Japanese workers in terms of muscle strength and physical load is necessary. Another method of determining weight limits is based on gender and handling positions, similar to the HSE guidelines. This method simplifies the determination of weight limits by considering only gender, height at which the load is handled, and distance from the body, allowing for easy representation in a straightforward table for intuitive use in the field. However, this method does not account for twisting movements, difficulty in handling the load, task frequency, or task duration. The weight limits will be further reduced when considering these factors. Further studies are needed to determine whether the current weight limits for United Kingdom workers (i.e., 25 kg for males and 16 kg for females) are appropriate for Japanese workers in terms of muscle strength and physical load. Given that the maximum weight capacities measured in this study tend to be lower than the weight limits set by the HSE guidelines, adjusting these limits for Japanese workers is necessary. The two methods, the equation from the ISO guidelines and the modified HSE guidelines, have different advantages, and their usefulness varies depending on whether detailed assessment or simplicity is prioritized. Therefore, it is necessary to consider which information to provide in different usage scenarios.

This study has several limitations. First, the IVD-CF obtained in this study was based on simulations rather than actual measurements. Obtaining actual measurements would require highly invasive methods, such as implanting sensors into the intervertebral discs, making the simulation-based method the most practical. The musculoskeletal simulation software used in this study has been validated for accuracy [13–15] and employed in previous studies [16–18]. Second, body weight changes can affect the skeletal structure and lifting posture. Therefore, using the same motion data for simulations across different weight levels may reduce the accuracy of the results. This poses a limitation in the simulations conditioned by body weight categories. To address this issue, this study included participants with different heights and body weights. Third, intervertebral disc strength may vary between individuals. Thus, the 3400 N threshold may not be applicable to all individuals, as addressed in a previous study [11]. Fourth, the generalizability of the present findings may be limited, as this study focused solely on Japanese workers. Differences in physical characteristics, such as body proportions and muscle mass, may influence lower back strain, potentially limiting the applicability of the results to other populations. Fifth, although this study proposed adopting the ISO guidelines and the modified HSE guidelines, its scope was limited and it did not present load values tailored to the Japanese population. It is necessary to assess whether the recommended maximum weights and multipliers are appropriate for Japanese workers. It is also important to consider additional factors not included in the present simulations, such as task frequency and duration, as these are known to influence the risk of low back pain.

 

## Conclusion

This study confirmed that load weight limits based on body weight percentages are inadequate for lumbar protection. Additionally, it revealed the shortcomings of this method. Particularly, the load weight limits set at 40% of body weight for males and 24% for females did not keep the lower back strain within acceptable limits. As body weight increases, the strain on the lower back also increases, leading to excessive lumbar strain. Moreover, tasks like handling at low positions or involving movement can significantly affect the strain on the lower back, exacerbating the issue. Therefore, the method of setting load weight limits based only on body weight percentages does not adequately protect the lower back and has inherent shortcomings.

Load weight limits should be determined based on the nature of the task rather than the body weight of the workers to better control strain. Although the equation from the ISO guidelines and the modified HSE guidelines are two suggested alternative methods, future studies should determine optimal values that consider muscle strength and body strain of Japanese workers. Additionally, each method has different advantages, and their use should be evaluated based on specific scenarios.

## Supporting information

**S1 Data. Maximum weight capacity.** Weight, maximum weight capacity (kg). Height: 1, shin; 2, knee; 3, waist; 4, chest; 5, head. Distance: N, near; F, far.
(CSV)

**S2 Data. Maximum L5-S1 IVD-CF.** IVD-CF, compressive force on the L5-S1 intervertebral disc; BW, body weight set in the simulation; Force, maximum L5-S1 IVD-CF (N). Height: 1, shin; 2, knee; 3, waist; 4, chest; 5, head. Distance: N, near; F, far.
(CSV)

## Acknowledgments

The authors would like to express their gratitude to Junko Fujita, Itoyo Komori, Maki Inoue, and Shuntaro Nakauchi for their assistance with the experiments and data analysis.

## Author contributions

**Conceptualization:** Fuyuki Oyama, Tanghuizi Du, Kazuyuki Iwakiri.

**Data curation:** Fuyuki Oyama, Tanghuizi Du.

**Formal analysis:** Fuyuki Oyama, Tanghuizi Du.

**Funding acquisition:** Kazuyuki Iwakiri.

**Methodology:** Fuyuki Oyama, Tanghuizi Du, Kazuyuki Iwakiri.

**Project administration:** Tanghuizi Du.

**Software:** Fuyuki Oyama.

**Supervision:** Kazuyuki Iwakiri.

**Writing – original draft:** Fuyuki Oyama.

**Writing – review & editing:** Fuyuki Oyama, Tanghuizi Du, Kazuyuki Iwakiri.

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
