## [Decision Letter · Decision Letter 0]

PONE-D-25-08962Inadequate lumbar protection with load weight limits based on body weight percentages: An experimental and simulation study of the weight limits set by the Japanese guidelines for preventing low back painPLOS ONE

Dear Dr. Oyama,

Thank you for submitting your manuscript to PLOS ONE. After careful consideration, we feel that it has merit but does not fully meet PLOS ONE’s publication criteria as it currently stands. Therefore, we invite you to submit a revised version of the manuscript that addresses the points raised during the review process. Both reviewers found the manuscript technically sound, with a clear structure, appropriate statistical analyses, and well-reported data. The study is notable for demonstrating that current guidelines may not provide sufficient lumbar protection, making it a relevant and timely contribution to the field. However, the manuscript requires important revisions before publication, focusing on the clarity of the results, expanded discussion, improvement of the conclusion, and emphasis on key findings. Specifically:

The presentation of results needs to be clearer, particularly by adding summary tables.The discussion section should go beyond simply restating the results. It needs to offer more analytical depth and clearer implications for guideline development.The conclusion currently does not provide critical reflection and needs improved articulation of the significance of the study, suggesting possible future directions.The authors should emphasise the central finding that weight limits based solely on body weight percentages do not adequately prevent lumbar strain.

Given these points, the manuscript must undergo major revisions, addressing all reviewer comments.

We look forward to receiving your revised manuscript.

Kind regards,

Massimo Brambilla, Ph.D.

Academic Editor

PLOS ONE

Journal Requirements:

2. Thank you for stating the following in the Acknowledgments Section of your manuscript: [This study was supported by the National Institute of Occupational Safety and Health, Japan (N-P03-01). The authors would like to express their gratitude to Junko Fujita, Itoyo Komori, Maki Inoue, and Shuntaro Nakauchi for their assistance with the experiments and data analysis.]

Please remove any funding-related text from the manuscript and let us know how you would like to update your Funding Statement. Currently, your Funding Statement reads as follows: “This study was supported by the National Institute of Occupational Safety and Health, Japan (N-P03-01). The funders had no role in study design, data collection and analysis, decision to publish, or preparation of the manuscript.”

Reviewers' comments:

Reviewer's Responses to Questions

**Comments to the Author**

1. Is the manuscript technically sound, and do the data support the conclusions?

Reviewer #1: Yes

Reviewer #2: Yes

2. Has the statistical analysis been performed appropriately and rigorously? 

Reviewer #1: Yes

Reviewer #2: Yes

3. Have the authors made all data underlying the findings in their manuscript fully available?

Reviewer #1: Yes

Reviewer #2: Yes

4. Is the manuscript presented in an intelligible fashion and written in standard English?

Reviewer #1: Yes

Reviewer #2: Yes

5. Review Comments to the Author

Reviewer #1: The manuscript is well organized, but some minor revisions are needed:

Lines 45-47: the values given refer to the best conditions for manual handling of loads. It is appropriate to specify this in the text.

Lines 56-60: perhaps the frequency of movements and the duration of the tasks are not considered either? If so, it is appropriate to specify this.

Lines 102-103: it is necessary to add more information regarding the ease of gripping the load: were there handles or not? It would be useful to add a photo or a figure.

Lines 330-331: the frequency of movements and the duration of the task were not considered either (factors that certainly influence the risk).

Reviewer #2: Japanese guidelines recommend that the load weight of manual handling should not exceed 40% of the body weight for males and 24% for females. The aim of the study was to identify potential shortcomings of the method of determining load weight limits based on body weight.

Here is my comment about it.

Methods:

In general a graphical representation of the various positions to be performed by the participants could help the understanding of the experimental method that was done. As well as a graphical description of the 3d motion analysis could be useful.

Results:

lines 223-268: a summary table could improve the understanding of the results.

Discussion:

lines 271-300: this whole part seems like a repetition of the results.

Give more support for the design of new guidelines that really prevent the problem of low back pain.

Conclusion:

The conclusions lack critical sense of the work and future developments are absent.

6. PLOS authors have the option to publish the peer review history of their article (what does this mean?). If published, this will include your full peer review and any attached files.

Reviewer #1: **Yes: **Massimo Cecchini

Reviewer #2: No

---

## [Author Response · Author response to Decision Letter 1]

8 May 2025

Academic Editor:

Thank you for considering our submission. Below are point-by-point responses to the issues that were raised.

1. The presentation of results needs to be clearer, particularly by adding summary tables.

-> We have added a summary table (Table 4) to enhance the clarity of the presented results.

2. The discussion section should go beyond simply restating the results. It needs to offer more analytical depth and clearer implications for guideline development.

-> We have reorganized the Discussion section and included a description of future guideline development incorporating the experimental findings.

3. The conclusion currently does not provide critical reflection and needs improved articulation of the significance of the study, suggesting possible future directions.

-> We have revised the Conclusion section to clearly state the key findings that address the objectives of this study. Additionally, we have added a description of future directions.

4. The authors should emphasise the central finding that weight limits based solely on body weight percentages do not adequately prevent lumbar strain.

-> We have revised the Conclusion section and emphasized the key finding that load weight limits based on body weight percentages are inadequate for lumbar protection.

Reviewer #1:

Thank you for reviewing our manuscript. Below are the responses to the comments.

1. Lines 45-47: the values given refer to the best conditions for manual handling of loads. It is appropriate to specify this in the text.

-> We have revised the description of the International Organization for Standardization (ISO) guidelines and clarified that the values are “the maximum recommended load weight under optimal conditions” (Lines 44–45).

2. Lines 56-60: perhaps the frequency of movements and the duration of the tasks are not considered either? If so, it is appropriate to specify this.

-> The Ministry of Health, Labor and Welfare of Japan (MHLW) guidelines do not set load weight limits based on load handling positions, frequency of movements, or duration of tasks. We have revised the description of the MHLW guidelines (Lines 57–60).

3. Lines 102-103: it is necessary to add more information regarding the ease of gripping the load: were there handles or not? It would be useful to add a photo or a figure.

-> As mentioned in lines 102–103, the load was a container with handles on both sides. We have added Fig 2, which shows the container and metal weights.

4. Lines 330-331: the frequency of movements and the duration of the task were not considered either (factors that certainly influence the risk).

-> We have clarified that the frequency of movements and duration of the task were not considered for the weight limits and included these factors in the paragraph where the shortcomings of MHLW guidelines are discussed (Lines 310–313).

Reviewer #2:

Thank you for reviewing our manuscript. Below are the responses to the comments.

1. Methods:

In general a graphical representation of the various positions to be performed by the participants could help the understanding of the experimental method that was done. As well as a graphical description of the 3d motion analysis could be useful.

-> We have added Fig 3, which shows various positions performed by the participants and examples of the captured motion data (Lines 125–128).

2. Results:

lines 223-268: a summary table could improve the understanding of the results.

-> We have added ANOVA summary tables (Table 4) and removed the numerical description of the ANOVA results from the main text (Lines 222–264).

3. Discussion:

lines 271-300: this whole part seems like a repetition of the results.

Give more support for the design of new guidelines that really prevent the problem of low back pain.

-> We have removed this part and provided more support for the future development of new guidelines (Lines 314–342).

4. Conclusion:

The conclusions lack critical sense of the work and future developments are absent.

-> We have highlighted the valuable insights gained from this study in the first paragraph of the Conclusion section (Lines 356–363). Additionally, we have included descriptions for future development in the second paragraph (Lines 364–369).

---

## [Decision Letter · Decision Letter 1]

PONE-D-25-08962R1Inadequate lumbar protection with load weight limits based on body weight percentages: an experimental and simulation study of the weight limits set by the Japanese guidelines for preventing low back painPLOS ONE

Dear Dr. Oyama,

Thank you for submitting your manuscript to PLOS ONE. After careful consideration, we feel that it has merit but does not fully meet PLOS ONE’s publication criteria as it currently stands. Therefore, we invite you to submit a revised version of the manuscript that addresses the points raised during the review process.

We look forward to receiving your revised manuscript.

Kind regards,

Woo-Keun Kwon

Academic Editor

PLOS ONE

Journal Requirements:

**Additional Editor Comments:**

Dear Authors,

Thank you for the opportunity to review your revised manuscript titled "Inadequate lumbar protection with load weight limits based on body weight percentages: an experimental and simulation study of the weight limits set by the Japanese guidelines for preventing low back pain." I commend the clarity of your analyses and the important contribution this study makes to occupational health and ergonomics.

Your findings provide convincing evidence that body weight–based load limits may not offer sufficient protection against lumbar strain, particularly under varied postural and lifting conditions. The integration of biomechanical simulation with task-specific data is a clear strength of this work.

That said, I believe the manuscript would benefit from a more thorough treatment of its methodological and contextual limitations. Specifically, I suggest adding a clearly labeled “Limitations” section (or expanding the existing discussion) to address the following points:

The use of uniform motion capture data across simulated body weights, which may not reflect actual kinematic variations.The omission of task duration and frequency—factors known to influence low back pain risk.The application of a fixed 3400 N IVD-CF threshold, despite known individual variability in lumbar spine tolerance.The limited generalizability of the findings beyond the Japanese working population.The practical challenges of implementing alternative guideline methods, and the absence of a proposed framework tailored to Japan.

Acknowledging these issues directly will enhance the interpretability, rigor, and credibility of your study. I believe these additions will strengthen the manuscript and support its value to both researchers and policymakers.

Sincerely,

Reviewers' comments:

Reviewer's Responses to Questions

**Comments to the Author**

1. If the authors have adequately addressed your comments raised in a previous round of review and you feel that this manuscript is now acceptable for publication, you may indicate that here to bypass the “Comments to the Author” section, enter your conflict of interest statement in the “Confidential to Editor” section, and submit your "Accept" recommendation.

Reviewer #1: All comments have been addressed

2. Is the manuscript technically sound, and do the data support the conclusions?

Reviewer #1: Yes

3. Has the statistical analysis been performed appropriately and rigorously? 

Reviewer #1: Yes

4. Have the authors made all data underlying the findings in their manuscript fully available?

Reviewer #1: Yes

5. Is the manuscript presented in an intelligible fashion and written in standard English?

Reviewer #1: Yes

6. Review Comments to the Author

Reviewer #1: The authors have submitted a revised version of the article, highlighting the changes made in response to the reviewers' suggestions. All comments have been addressed. The article can now be published.

7. PLOS authors have the option to publish the peer review history of their article (what does this mean?). If published, this will include your full peer review and any attached files.

Reviewer #1: **Yes: **Massimo Cecchini

---

## [Author Response · Author response to Decision Letter 2]

4 Jun 2025

Academic Editor:

Thank you for considering our submission. Below are the responses to the issues that were raised.

1. That said, I believe the manuscript would benefit from a more thorough treatment of its methodological and contextual limitations. Specifically, I suggest adding a clearly labeled “Limitations” section (or expanding the existing discussion) to address the following points:

• The use of uniform motion capture data across simulated body weights, which may not reflect actual kinematic variations.

• The omission of task duration and frequency—factors known to influence low back pain risk.

• The application of a fixed 3400 N IVD-CF threshold, despite known individual variability in lumbar spine tolerance.

• The limited generalizability of the findings beyond the Japanese working population.

• The practical challenges of implementing alternative guideline methods, and the absence of a proposed framework tailored to Japan.

Acknowledging these issues directly will enhance the interpretability, rigor, and credibility of your study. I believe these additions will strengthen the manuscript and support its value to both researchers and policymakers.

→ In response to your comments, we have added the relevant limitations to the Discussion section.

Journal Requirements:

→ We appreciate your bringing the inaccuracies in our reference list to our attention. In response, we have conducted a thorough review to ensure that all references are accurate and complete. We have also verified that no retracted publications are included. Furthermore, we have made the following revisions to the citations:

[3]: Replaced the URL to directly link to the cited document file.

[5]: Corrected the country name from “England” to “United Kingdom”.

[6]: Replaced the citation with the book titled “2023 TLVs and BEIs”.

[19]: Added “(in Japanese)” to the end of the title, replaced the URL to directly link to the cited document file, and included the access date.

---

## [Editor Report · Decision Letter 2]

Inadequate lumbar protection with load weight limits based on body weight percentages: an experimental and simulation study of the weight limits set by the Japanese guidelines for preventing low back pain

PONE-D-25-08962R2

Dear Dr. %Fuyuki Oyama%,

We’re pleased to inform you that your manuscript has been judged scientifically suitable for publication and will be formally accepted for publication once it meets all outstanding technical requirements.

Kind regards,

Woo-Keun Kwon

Academic Editor

PLOS ONE

Additional Editor Comments (optional):

As a handling editor of this manuscript, I have carefully examined your revised submission and found that you have sincerely and thoroughly addressed all the comments and suggestions provided in the previous review. Your thoughtful revisions have significantly enhanced the clarity, scientific rigor, and overall quality of the manuscript.

I appreciate your diligence and responsiveness throughout the review process, and I am confident that your work will make a valuable contribution to the literature and be of interest to our readers.
---

## [Editor Report · Acceptance letter]

PONE-D-25-08962R2

PLOS ONE

Dear Dr. Oyama,

I'm pleased to inform you that your manuscript has been deemed suitable for publication in PLOS ONE. Congratulations! Your manuscript is now being handed over to our production team.

Kind regards,

on behalf of

Dr. Woo-Keun Kwon

Academic Editor

PLOS ONE